# Randomised controlled trial of a person-centred transition programme for adolescents with type 1 diabetes (STEPSTONES-DIAB): a study protocol

Anna Lena Brorsson [1,2] Ewa-Lena Bratt,[3,4] Philip Moons,[3,5] Anna Ek,[6,7] Elisabeth Jelleryd,[7] Torun Torbjörnsdotter,[7] Carina Sparud-Lundin[2,3]

For numbered affiliations see end of article.

**Correspondence to**
Dr Anna Lena Brorsson;
anna-lena.brorsson@ki.se

## ABSTRACT

**Introduction** Adolescence is a critical period for youths with chronic conditions, when they are supposed to take over the responsibility for their health. Type 1 diabetes (T1D) is one of the most common chronic conditions in childhood and inadequate self-management increases the risk of short-term and long-term complications. There is a lack of evidence regarding the effectiveness of transition programmes. As a part of the Swedish Transition Effects Project Supporting Teenagers with chrONic mEdical conditionS research programme, the objective of this study is to evaluate the effectiveness and experiences of different transitional care models, including a person-centred transition programme aiming to empower adolescents with T1D to become active partners in their health and care.

**Methods and analysis** In this randomised controlled trial, patients are recruited from two paediatric diabetes clinics at the age of 16 years. Patients are randomly assigned to either the intervention group (n=70) where they will receive usual care plus the structured transition programme, or to the control group (n=70) where they will only receive usual care. Data will be collected at 16, 17 and 18.5 years of age. In a later stage, the intervention group will be compared with adolescents in a dedicated youth clinic in a third setting. The primary outcome is patient empowerment. Secondary outcomes include generic, diabetes-specific and transfer-specific variables.

**Ethics and dissemination** The study has been approved by the Ethical Review Board in Stockholm (Dnr 2018/1725-31). Findings will be reported following the Consolidated Standards of Reporting Trials statement and disseminated in peer-reviewed journals and at international conferences.

**Trial registration number** NCT03994536

### Strengths and limitations of this study

► The evidence of transition interventions targeting different long-term conditions is still insufficient; this study will expand current knowledge on transitional care.
► Experience from an ongoing project where the person-centred transition programme is evaluated in another group of adolescents with long-term conditions assures study fidelity.
► Blinding of participants is not possible.
► Long-term follow-up of the effectiveness of transition interventions is important and desirable. Although not included in this study, prolonged follow-up is needed.

will comprise a shift in care when adolescents are transferred from a paediatric to adult care services. Simultaneously, there is a shift in roles between parents and adolescents, where the adolescent needs to start taking over the responsibility for their health and care. Furthermore, this shift occurs during a vulnerable developmental period, when adolescents are exposed to unique challenges.[2 3] A smooth transition to adulthood and well-timed and planned transfer to adult care allows adolescents to optimise their ability to assume adult roles and functioning, which can improve medical outcome.[4] In this article, transfer is defined as the actual event when adolescents move from a paediatric to adult care services, while transition is defined as the process by which they become prepared to take charge of their lives and their health in adulthood.[5]

Transition programmes for adolescents with chronic conditions aim to prepare and support them during the transition to adulthood and transfer to adult care so that they can gradually become active partners in their health and care. In this respect, adequate

## INTRODUCTION

Worldwide, nearly 132 600 000 children and adolescents under 20 years develop type 1 diabetes (T1D) each year. The annual mean incidence rate varies from 0.1–60 to 100 000, with Sweden as one of the 'top' incidence countries.[1] Children with T1D need lifelong medical follow-up to prevent acute and long-term complications. The lifelong follow-up

self-management and participation in care are essential and it is of utmost importance that patients feel empowered.[6] Patient empowerment is a critical attribute of person-centred care (PCC) and highlights the importance of knowing the person behind the patient.[7] Patient empowerment is considered to be 'An enabling process or outcome arising from communication with the healthcare professional and a mutual sharing of resources over information relating to illness, which enhances the patient's feelings of control, self-efficacy, coping abilities and ability to achieve change over their condition'.[8] Research on the application of patient empowerment in adolescent health and transitional care is limited. The application of PCC principles in young persons with chronic diseases is unprecedented, although there is a growing demand for PCC in diabetes care. Moreover, research on transition and transfer and on interventions that have the potential to improve both processes is important. Nevertheless, evidence on the effectiveness of transition programmes using a randomised controlled trial (RCT) is sparingly reported.[9] Evidence base for interventions that improve outcomes for this specific target group is limited and not adequate in guiding clinical practice to support better self-management and outcomes in youths with T1D.[10]

It is known from previous studies that transfer from paediatric to adult care services is associated with deterioration in health of adolescents with certain chronic conditions.[9] For adolescents with T1D, this includes non-optimal glycaemic control and increased diabetes related hospitalisations after transfer, as well as general patient dissatisfaction with the transition experience. Poor clinic attendance after transfer is considered a risk factor.[11 12] Potential importance of continuity of care, support, education and individualised support is highlighted across different types of interventions targeting young persons with T1D.[13] Within diabetes care, healthcare professionals agree that a lack of an effective and structured transfer from paediatric to adult care may contribute to insufficient continuity of healthcare and increased risk of adverse outcomes in young adults with T1D.[14 15]

To investigate the effectiveness of a structured person-centred transition programme for adolescents with chronic conditions, the Swedish Transition Effects Project Supporting Teenagers with chrONic mEdical conditionS (STEPSTONES) has been established. STEPSTONES is a research project in which a transition programme has been developed and currently tested in an ongoing RCT in adolescents with congenital heart disease (CHD).[16 17] Theoretical framework for the transition programme is supported by the conceptualisation of empowerment defined by Small et al[8] and with other attributes related to the concept. The components are following the philosophy of PCC,[7] operationalised and guided by several change theories/models described by Acuña Mora et al.[18] The STEPSTONES transition programme is generic in nature and can therefore be expanded to other childhood-onset conditions, such as T1D.

## Objectives and hypothesis

The overall objective of STEPSTONES-DIAB is to evaluate the effectiveness of transitional care, aiming to empower adolescents with T1D to become active partners in their health and care.

The following hypotheses will be tested in two steps:
1. Adolescents with T1D who receive a structured, person-centred transition programme have a higher patient empowerment score than adolescents who receive usual care.
2. A structured, person-centred transition programme is equivalent to a dedicated youth clinic in empowering adolescents with T1D.

## METHODS AND ANALYSIS

We have used the recommendations in the Standard Protocol Items: Recommendations for Interventional Trials reporting guidelines in this protocol paper.[19]

### Design and setting

An RCT is being conducted in two paediatric diabetes clinics in which patients are randomly assigned to either the intervention group (study arm 1) where they receive usual care plus the structured transition programme, or to the control group (study arm 2) where they receive usual care only. In parallel, all patients followed up in a special youth clinic comprise study arm 3, to be compared with study arm 1 at a later stage. All the clinics are located at University Hospitals in Stockholm, Sweden. The number of transferred patients per year differs between the centres. For an overview of the three study arms, see figure 1.

### Participants and recruitment

Inclusion criteria for the study:
► Literate, Swedish-speaking adolescents with T1D.
► Aged 16 years.
► Diabetes duration >1 year.
► If other diagnoses are present, T1D must be the primary diagnosis and not a side effect of other diagnoses or treatment for example, cystic fibrosis or cortisone-triggered diabetes.

Parents/guardians, as well as the adolescents, will be asked to participate, the inclusion criterion being that they are literate and Swedish speaking. Participants will be excluded if they are diagnosed with conditions affecting cognitive abilities and may have difficulties understanding and/or completing questionnaires.

A transition coordinator (TC) will contact 16-year-old adolescents eligible for inclusion in the RCT (study arm 1 and 2) during scheduled outpatient visits or by telephone. The adolescents and their guardians will be informed both verbally and in writing about the purpose of the study, its voluntary nature, the right to cease participation at any time, the data storage procedure and the strict anonymous processing of data. In the youth clinic (study arm 3), a data collection officer (DCO) will handle recruitment and data collection.

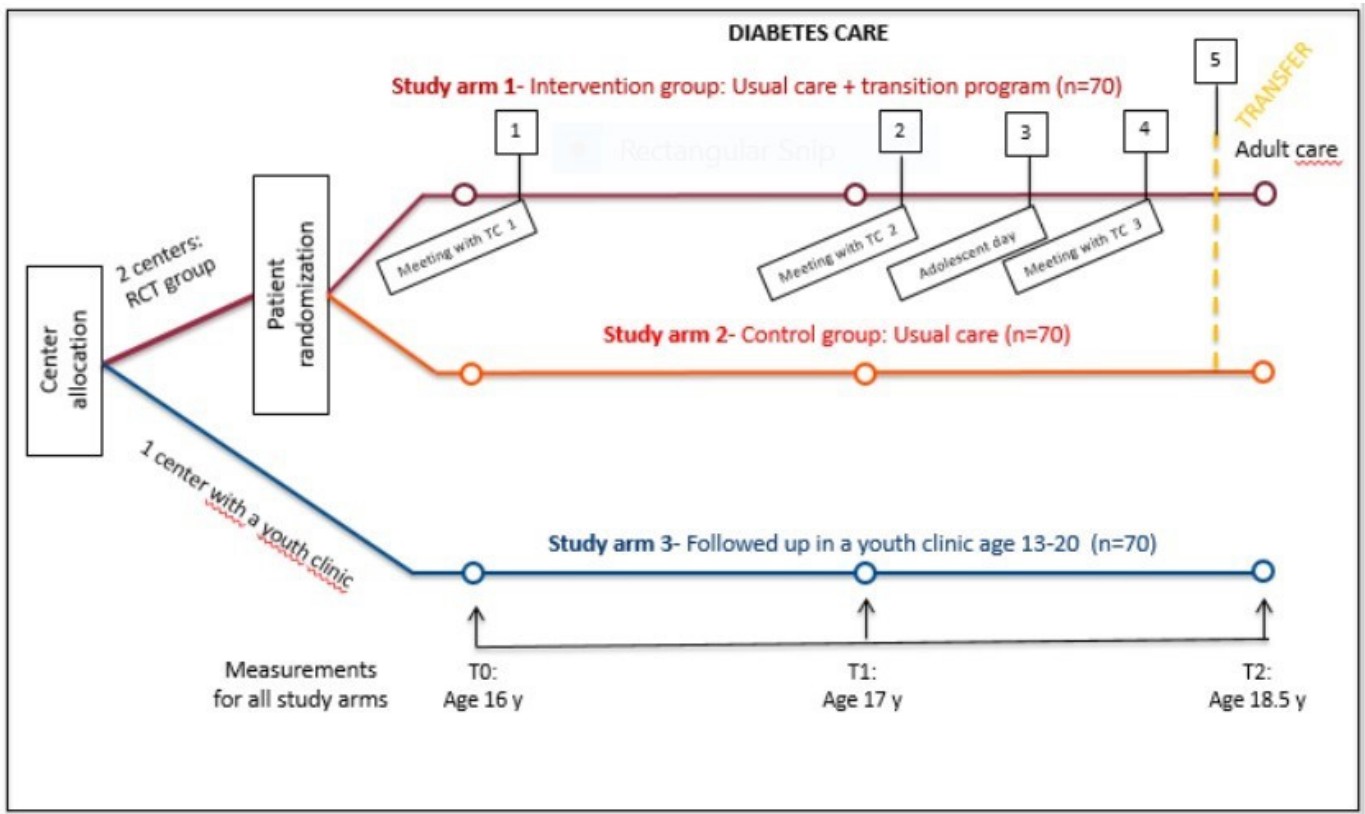

**Figure 1** Overview of the study design. RCT, randomised controlled trial; TC, transition coordinator.

At both RCT centres, participants are assigned to either the intervention or the control group. This is done with a web-based randomisation system (1:1) employing stratified block randomisation with a random variable block size. Block randomisation ensures that the TCs in the two centres have relatively continuous exposure to the intervention over time, allowing them to keep their skills up to date and to ensure the fidelity of the intervention. The fact that the block sizes vary randomly minimises the possibility of the TCs being able to predict group assignments. Stratification is made by the centre to achieve an equal distribution of patients in the intervention, with control groups at both centres. Furthermore, stratification for sex and glycated haemoglobin (HbA1c) is crucial to ensure the distribution in the groups due to the knowledge that female adolescents have poorer glycaemic control compared with males, and that adolescents with unsatisfactory glycaemic control are a high-risk population.[20 21] The grouping for HbA1c is based on ISPAD (International Society for Pediatric and Adolescent Diabetes) guidelines 2014.[20] Overall, this approach decreases the within-centre variability and reduces the risk of bias and confounding.[22] An intention-to-treat principle will be used.

### Intervention

#### Transition programme (intervention group)

A multicomponent intervention, adapted from the brief transition programme of Hilderson et al[23] and the STEP-STONES-CHD project,[16] will be implemented. This transition programme comprises eight key components: (1) a TC; (2) a person-centred written transfer plan; (3) provision of information and education about T1D and treatment, health behaviours, dealing with school, friends, leisure activities and the importance of treatment adherence; (4) high access to the TC via telephone/sms; (5) information about and contact with the adult diabetes care team; (6) meeting with other young people with T1D; (7) support for and guidance of parents and (8) the actual transfer to the adult diabetes outpatient clinic. The components are implemented in five steps: (1) first visit with TC at the diabetes paediatric outpatient clinic at the age of 16 years; (2) second visit with TC at the diabetes paediatric outpatient clinic at the age of 17 years; (3) 'adolescent day' group information with parents and adolescents, with the participation of healthcare professionals from the adult outpatient clinic and young adults with T1D; (4) third visit with TC at the diabetes outpatient clinic at the age of 18 years and (5) the current transfer to one of the eight adult outpatient clinics that care for adults with T1D in the area (figure 1). The intervention is delivered by specialised trained paediatric nurses at the two paediatric diabetes outpatient clinics (study arm 1 and 2). The content of the patient education is adapted to T1D.

#### Usual care (intervention and control group)

The current usual care is provided to all participants, irrespective of whether they are randomised to the intervention or not. Usual care includes check-up visits to the paediatric diabetes outpatient clinic according to

international, national and local guidelines every third months, with adjustments for individual needs.[24 25] Visits include a meeting at the paediatric diabetes outpatient clinic with a nurse or a physician and it is up to each family if the parents attend these visits. In the area, there are several adult diabetes outpatients' clinics to be transferred to. In one of the two intervention clinics usual care includes that adolescents planned to be transferred to the adult diabetes outpatient clinic in the same hospital will be scheduled for a transfer visit in groups. If an adolescent does not participate in this group referral, the same routines apply as for the rest of the adolescents in the control group, that is, they will be referred directly to the chosen adult diabetes clinic. None of the two intervention centres has a formal TC in place and no additional transfer preparation.

### Youth clinic

At the dedicated youth clinic (study arm 3), adolescents with T1D will be followed up from 13 years of age and a more flexible transfer to adult care at 19–20 years of age will be applied. Usual care in this clinic includes visits to the diabetes nurse or the physician at the unit every second to third month with closer visits if needed, according to the patient's glycaemic control. Midwives and social workers are employed at the unit and, if needed, visits to these can be arranged. The midwives provide advice on birth control, STDs (sexually transmitted diseases) and consultations in case of pregnancy. In the case of medical abortion, this can be performed in the youth clinic in a youth-friendly environment. No formal TC is in place, nor a formalised transition/transfer plan.

### Intervention fidelity

The special training of TC together with their experience of working with children and adolescents with T1D will ensure compliance that the components are implemented in line with the developed intervention. Prior to the start of this study, the TCs received 5 days of training including: adolescent psychosocial development and health with focus on chronic diseases in general and T1D in particular; PCC and empowerment; common neuropsychiatric diagnoses; the legal and human rights of young people. Moreover, they received training in communicating and interviewing young people using the psychosocial interview guide HEEEADS, the acronym standing for home, education/employment, activities, drugs, sexuality and suicide/depression. The method uses structured questions to facilitate communication and create a sympathetic, confidential, respectful environment.[26] The TC training was provided by experts within each field. TCs were also introduced to the administrative parts of the study in line with Good Clinical Practice (GCP) and have participated in a course in GCP.

Standard operating procedures (SOPs) were developed to ensure consistency throughout the study. These SOPs include ethical approval, study preparation and study administration, participant recruitment and survey completion; usual care, data storage, entry and security, study progress, process evaluation and publication policy. The Medical Research Council guidance on process evaluation of complex interventions is applied to monitor intervention fidelity.[27] Acceptability, adherence/fidelity and attrition are monitored using specific assessment forms; enrolment/follow-up form, and intervention implementation form to secure intervention delivery. These are filled out by the TCs during the entire intervention period. Qualitative interviews after completed delivery of the intervention will provide insights into the context of the intervention and the mechanisms of impact, based on the experiences of participants.

To avoid contamination, the TCs will not care for adolescents from the control group or adolescents eligible for the study, and team members at the outpatient clinic who are not involved in the study will only be given a brief outline of the intervention programme content.

### Data collection

Baseline data will be collected when participants (the adolescent and one or two parents) have agreed to participate in the study. Participants are asked to complete the set of pen-and-paper questionnaires at 16, 17 and 18.5 years of age (figure 1). These are sent by post to the participants' home address before the next scheduled outpatient visit. The pack contains an instruction letter, a set of questionnaires and a prepaid and addressed reply envelope. Baseline data (T0) will be collected before any intervention is started (study arm 1). For T0, the most recent HbA1c value before inclusion in the study will be used. It is estimated that the questionnaires will take approximately 30 min to complete. Participants will be asked to complete the set of questionnaires themselves and to return them within 3 weeks. If the questionnaires are not returned within this time frame, patients will be reminded by a new set of questionnaires or asked to complete them while waiting in the waiting room at the next visit.[16 28]

### Outcome measures

The questionnaire obtained at T0 provides background information including: sex, age, education level, born in Sweden or not; adolescent/parents born in Sweden or not; living situation (adolescent)/marital status (guardian); number and order of siblings. For guardians, employment is also required. Person-reported outcome measures are linked to empowerment and PCC by evaluating the personal determinants of generic and illness specific nature (knowledge, self-efficacy and self-management skills).[18]

### Primary outcome measures
#### Empowerment

The primary outcome is empowerment. Empowerment is measured with the Gothenburg Young Persons Empowerment Scale (GYPES).[29] This scale has been tested in a cross-sectional study in order to determine

its psychometric properties in adolescents with CHD and T1D and supporting GYPES validity and reliability as a tool for assessing young persons' empowerment.[26] The five subscales measure: (1) self-perceived level of understanding of their disease (knowledge and understanding); (2) the capacity patients have to handle their disease (personal control); (3) the effect their illness has on their lives and sense of self (identity); (4) the capacity to make decisions along with the healthcare professional (shared decision-making); and (5) the ability to share their experiences and help others who are going through a similar situation (enabling others). The primary outcome, analysed at T2, is the total score ranging from 15 to 75 points, with a higher score reflecting a higher level of empowerment.[29]

### Secondary outcome measures

The secondary outcome variables chosen are other important developmental tasks in adolescence and emerging adulthood and in relation to T1D. When evaluating interventions aimed at improving the outcome of young adults with T1D, some recommended core outcomes are set.[30] These eight outcomes are: measures of diabetes-related burden or stress, diabetes-related quality of life (QoL), number of severe hypoglycaemic events, self-management behaviour; number of instances of diabetic ketoacidosis (DKA), objectively measured HbA1C, level of clinic engagement and perceived level of control over diabetes. In addition, we measure

indicators for successful transition and transfer outcomes as proposed by Coyne et al,[31] which include clinic attendance, hospitalisation rates, disease-specific outcomes and transfer-specific satisfaction. See table 1 for an overview of included variables and measurements. Secondary outcomes include endpoint at T2 and change from baseline to T2 (also empowerment), while the subscales in questionnaires will be included in exploratory analyses.

### Generic

Health behaviours are measured with the Health Behaviour Scale. This questionnaire has been validated in adolescents with CHD but is also considered to be valid for the diabetes group. Health behaviours are activities a person undertakes to maintain or improve health and prevent diseases. This scale assesses alcohol consumption, tobacco use, dental care and physical activity. The scale has 15 items that help to calculate three summary risk scores: substance use (score 0–100), dental hygiene (score 0–100) and total health risk score (score 0–100). Higher risk scores represent unhealthier behaviours.[32]

### Diabetes specific

Diabetes duration, insulin administration (multiple daily injections or continuous subcutaneous insulin infusion), blood glucose testing (self-monitoring blood glucose testing/continuous glucose monitoring, CGM/intermittent scanning CGM) are descriptive variables. Hypoglycaemic events and DKA will be summed up for the whole

**Table 1** Overview of variables and measurements in Stepstones-DIAB

| Variables/indicators | Measurement | Time point | Source |
|---|---|---|---|
| **Primary outcome** | | | |
| Patient empowerment | Gothenburg Young Persons Empowerment scale (DIAB)[29] | T0, T1, T2 | A |
| **Secondary outcomes** | | | |
| **Generic** | | | |
| Health behaviour | Health Behaviour Scale[32] | T0, T1, T2 | A |
| **Diabetes specific** | Diabetes duration<br>Insulin administration (MDI/CSII)<br>Blood glucose testing (SMBG/CGM/isCGM) | T0 | NDR |
| Glycaemic control | HbA1c, hypoglycaemic events, diabetic ketoacidosis | T0, T1, T2 | NDR |
| Diabetes burden | Check your health[33] | T0, T1, T2 | A |
| HRQoL/level of control | DisabKids Chronic Generic Measure-12[34–36]<br>Disabkids Diabetes Module-10[34 36 37] | T0, T1, T2 | A |
| **Transfer specific** | | | |
| Transition readiness | Readiness for Transition Questionnaire[38] | T0, T1, T2 | A+P |
| Uncertainty (parents) | Uncertainty Scale (Linear Analogue Scale)[39] | T0, T1, T2 | P |
| Clinical indicators | Clinic attendance rates, pretransfer and post-transfer Time between last visit in PC and first visit in AC | T2 | Admin |
| Transitional Care Experiences Questionnaire | In progress | T2 | A |

A, adolescents; AC, adult care; CGM, continuous glucose monitoring; CSII, continuous Ssubcutaneous Iinsulin Iinfusion; HbA1c, glycated haemoglobin; isCGM, intermittent scanning continuous glucose monitoring; MDI, multiple daily injection; NDR, National Diabetes Registry; p, parents; PC, paediatric care; SMBG, self-monitoring blood glucose testing.

study period and compared at T2. HbA1c will be analysed using DCA Vantage (Siemens Healthcare Diagnostics AB, Upplands Väsby, Sweden) and the test results collected from the National Diabetes Register. Analyses will be performed at T2 and change from baseline to T2.

'Check your Health' measures self-perceived physical and emotional health, social relationships and general QoL on four vertical thermometer scales ranging from 0 to 100, with 0 indicating low self-perceived health. Each scale indicates self-perceived health with diabetes and, on the same scale, self-perceived health imagined without diabetes. The measured difference between self-perceived health with and the imagined without diabetes is defined as the burden of diabetes.[33]

DISABKIDS Chronic Generic Measure-12 measures general QoL and the level of distress caused by a chronic disease. The short-form of the DISABKIDS chronic generic module consists of 12 five-point Likert-scaled items assigned to the three domains: mental, social and physical. Higher scores represent better outcome.[34–36]

DISABKIDS Diabetes Module consists of 13 five-point Likert scaled items and has two scales: an Impact and a Treatment scale. The Impact scale describes emotional reactions of the need to control everyday life and restrictions of one's diet. The Treatment scale refers to carrying equipment and planning treatment. Higher scores represent better outcome.[34 36 37]

### Transfer specific

The Readiness for Transition Questionnaire (adolescent and parent version) examines two aspects. First, the overall transition readiness is assessed, using two items ranging from 1 to 4. The sum of these two items results in a score ranging from 2 to 8. Second, the frequency of adolescent responsibility and parental involvement are reported for 10 different health behaviours on a four-point Likert scale. Each variable results in a total score ranging from 10 to 40, with higher scores indicating higher adolescent responsibility or parental involvement, respectively.[38]

Parental uncertainty is measured using a 10 cm Linear Analogue Scale (LAS) with the end points 'not uncertain at all' (=0) and 'extremely uncertain' (=100). Parental uncertainty comprehends parents' perceptions of certainty/uncertainty prior to their child's transfer to adult care. This LAS was developed for an ongoing study in the STEPSTONES-CHD project and has been used in a cross-sectional study.[39]

The new questionnaire on transfer and transition experiences is under construction and will be tested on young persons aged 19–21 years with long-term conditions and experience of the transfer to adult care in order to evaluate its psychometric properties.

Clinical indicators, such as attendance for scheduled outpatient visits before and after transfer to adult clinic and the duration between last visit in paediatric care and the first in adult care, will be collected from the medical record.

Information about validity, reliability and responsiveness of the questionnaires is provided in a online supplementary file.[29 32–35 37–40]

### Data management

The completed questionnaires are entered into a secure research database, using a web-based data entry system, accessible for research group only. Each patient is assigned a unique study code and this code will be used as a unique identifier in the database to ensure patient anonymity. The list of study codes and the list of eligible participants will be stored separately in a secure file to which only the TC coordinators have access. In study arm 3, a DCO will handle this process.

Data monitoring and quality checks will be undertaken by a team member with vast experience of database management. To make data collection procedures uniform, TCs and the DCO have been provided with thorough training; regular visits will be made by the project coordinator and regular quality assurance sessions will be organised for TCs and DCO.

Owing to the nature of the intervention and the study design, it is not possible to blind the participants but TCs in charge of carrying out the intervention and data collection are not involved in the preparation of the intervention design or in the statistical analyses.

### Proposed sample size

Based on the primary outcome, we target an improved patient empowerment score of 5.25 points on a scale from 15 to 75 (ie, 0.5 SD).[29] For two-sided tests with alpha=0.05 and power=80%, 63 patients are needed in each arm of the RCT. In order to compensate for a potential 10% dropout rate, 70 patients will be included in each arm. Given that the two paediatric diabetes outpatient clinics together have about 100 patients in each age cohort of adolescents, and considering a 70% participation rate, a 24-month recruitment period is needed to enrol 140 patients in study arm 1 and 2.

### Data analysis

The primary endpoint (total score of the level of empowerment at T2) will be analysed using Fisher's non-parametric permutation test unadjusted between the intervention group and the control group in the RCT on the Full Analysis Set with imputation of missing values using stochastic imputation. A sensitivity analysis will be conducted with analysis of covariance (ANCOVA) adjusting for stratification variables without imputation. If confounders are found, complementary analyses will be conducted using ANCOVA between the two groups adjusted for these confounders. A confounder is a baseline variable that differs between the groups and influences the primary outcome variable. For comparison between the two randomised groups, Fisher's non-parametric permutation test will be used for continuous variables, Fisher's exact test for dichotomous variables, Mantel-Haenszel $X^2$ test for ordered categorical variables

and Pearson $X^2$ test for non-ordered categorical variables. Missing data will be handled with stochastic imputation. Mean difference between the two groups with 95% CI will be presented for all continuous and dichotomous variables. For comparison within groups, Fisher's non-parametric permutation test for paired observation will be used for continuous variables and Sign test for dichotomous variables and ordered categorical variables.

All analyses will be predefined in a detailed statistical analysis plan before data base lock. In a later stage, a comparison between the intervention group and a youth clinic group (study arm 3) will also be performed. All significance tests will be two sided and conducted at the 5% significance level.

## Patient and public involvement

During the planning phase of this study, young persons from the Swedish patient organisation, 'Young Diabetes', have been involved as advisors in the development of certain parts of the intervention. Insufficient transitional care has been on their agenda for several years and the initiative to evaluate support for transition and transfer is therefore highly sought after. Representatives from Young Diabetes will continuously be involved as a reference group during the entire study to secure consideration of young peoples' priorities, experiences and preferences in all phases. These representatives also participate in the event 'Adolescent Day'. There is an established advisory board in STEPSTONES that has been involved in the development of the intervention, also including young persons with CHD and parent representatives.[16] Henceforth, this advisory board will serve the entire STEPSTONES project, and thus young persons with diabetes have been included in the advisory board. Further, this group includes experts on adolescent health and medicine, as well as on the research and implementation of PCC.

## Trial duration

Recruitment of participants began in August 2019 and is expected to be completed in August 2021. Data collection is planned to be completed in January 2024. When data collection is completed, the dataset will be closed and the analysis of data and the dissemination of the results will begin.

## Ethical considerations

For this study, Ethics approval has been obtained from the Regional Ethics Review Board in Stockholm (Dnr 2018/1725-31). A multicentre ethical application includes mandatory approvals from the head of the department at each participating centre.

Before inclusion, an information pack is given to the adolescents and their parents to explain the purpose of the study, its voluntary nature, the right to cease participation at any time, the data storage procedure, and the strict anonymous processing of data. Patients will only be included if the adolescent, who is a minor at the time

of inclusion and the parents/guardians provide written informed consent.

The risk of participating in this study is low, as the intervention is supportive in nature. If the transition programme is shown to be effective, the benefits may be substantial. We consider the participants to be vulnerable persons. For some participants, filling out the questionnaires may bring up some unpleasant memories and/or feelings, so contact details of the TCs/DCOs, as well as organisations which can help them cope with these feelings, will be provided in the information pack. Participants are covered by Swedish patient insurance in case of adverse events (AEs).

## Follow-up on AEs

All AEs will be reported within a week to the principal investigator and the project manager. In case of a serious AE, the report will be made immediately. Appropriate measures will be undertaken in every AE to ensure patent safety.

## Dissemination

The results from this study will be published in peer-reviewed journals and will follow the recommendations of the Consolidated Standards of Reporting Trials statement.[40] Further, abstracts for poster and oral presentation will be submitted to national and international conferences and findings communicated and discussed with healthcare and patient representatives for potential implementation.

## DISCUSSION

This study relies on the experiences from two previous research projects aiming to promote the transition process for youths. The first instance was for adolescents with rheumatoid arthritis, the second is the current STEPSTONES-CHD research project by testing a further developed transition programme in adolescents with CHD,[16 41] and now we are expanding to evaluate the effectiveness of a structured, person-centred transition programme for adolescents with T1D. The focus is to increase their level of empowerment in order to promote their ability to navigate through the adult care system, participate in care planning, and increase self-management and decision-making skills.[6 7]

As previously stated, creating effective clinical processes for the transition from paediatric to adult care is particularly important in order to optimise well-being and health in emerging adults with T1D. This includes achieving target glycaemic control to prevent long-term complications and to maximise lifelong functioning. Although there is a lack of proven strategies to achieve these goals, programmes that particularly target adolescents with T1D through education, skills training, special transition clinics and/or addition of TCs appear to be promising.[10 42] However, the current knowledge regarding this transition process calls for ongoing and expanding

research initiatives in order to determine the effectiveness of transition programmes for adolescents with different long-term conditions.[9 43]

One of the strengths of this study is the chosen design; employing an RCT is considered the gold standard when evaluating the effectiveness of interventions and provides the highest level of evidence. By using stratified block randomisation, we ensure that the intervention and control groups are balanced in terms of potential confounding factors, for example, equal distribution, thereby minimising the risk of biasing the result. In a previous empowerment-based study including a person-centred group education for youths with T1D, the results highlighted the importance of an approach that strengthens young people's independence and autonomy. The parents' attitude was also highlighted in this process. The adolescents described the importance of a permissive atmosphere and that it was perceived as positive to meet at new healthcare professional in these encounters. Moreover, the adolescents said it was empowering to meet others with T1D.[44] For HbA1c, an effect could only be seen for boys.[45] Another strength is that projects can use the lessons learnt from STEPSTONES-CHD in terms of feasibility and fidelity. For example, experiences of SOPs for recruitment and data collection will secure data entry. Moreover, in the training of the TCs, the experienced TCs from STEPSTONES-CHD could be used as tutors and mentors, which further secures delivery of the generic intervention components in STEPSTONES-DIAB. By employing experienced diabetes nurses as TCs, we ensure that the generic components are adapted to and relevant for diabetes care. Further, the generic components in the person-centred transition programme, as well as the generic outcome measures, provide opportunities to compare findings across different conditions. If the programme is proven effective, these circumstances may facilitate future implementation of transfer and transition strategies in clinical practice.

A challenge in complex interventions like transition programmes is the multiple interacting components and behaviours required by those delivering and receiving the intervention. To evaluate the full extent of a complex intervention in addition to effectiveness, the evaluation must incorporate process assessments of how the intervention is being delivered in practice and what causal mechanisms lead to the desired outcomes. Based on the experiences from the more extensive process evaluation undertaken in STEPSTONES-CHD,[17] we have identified certain process evaluation components that also need to be undertaken in STEPSTONES-DIAB in order to promote future implementation. These components reflect what is delivered (fidelity, adherence, dose, reach, acceptability) and are provided through participant observations, the intervention implementation form, enrolment form and transition plans.[17]

Some limitations need to be acknowledged in this study. First, this intervention focuses on preparation for transition to adulthood and the actual transfer to adult care. Additional interventions may be needed to secure continuity of care after transfer. Second, the lack of blinding procedures poses a limitation in that the nature of study makes blinding patients and TCs impossible. Third, the short follow-up period of 6 months at the age of 18.5 years cannot be considered enough to evaluate long-term effects of the transition programme. There is a great need to evaluate such effects[9] and these must be followed up in separate studies after additional consent and funding.

## CONCLUSION

The empirical underpinning of the guidelines, recommendations and statements regarding successful transitional strategies is weak and to be considered expert-based rather than evidence based. Based on an RCT design and building on previous projects, STEPSTONES-DIAB has the research capacity to provide high-level evidence within this area, and thereby contribute to filling the gaps in scientific knowledge related to transitional care of adolescents with T1D.

**Author affiliations**
[1]Department of Neurobiology, Care Sciences and Society, Karolinska Institutet, Stockholm, Sweden
[2]Gothenburg Centre for Person-Centred Care (GPCC), University of Gothenburg, Gothenburg, Sweden
[3]Institute of Health and Care Sciences, University of Gothenburg, Gothenburg, Sweden
[4]Department of Pediatric Cardiology, The Queen Silvia Children's Hospital, Gothenburg, Sweden
[5]Department of Public Health and Primary Care, KU Leuven, Leuven, Belgium
[6]Division of Pediatrics, Department of Clinical Science, Intervention and Technology, Karolinska Institutet, Stockholm, Sweden
[7]Department of Paediatric Diabetes, Astrid Lindgren's Children's Hospital, Karolinska University Hospital, Stockholm, Sweden

**Contributors** CS-L, E-LB, PM and ALB designed the study. ALB and CS-L drafted the manuscript. E-LB, PM, AE, EJ and TT participated in reviewing and editing. All authors read and approved the final manuscript.

**Funding** This work was supported by the Swedish Child Diabetes Foundation (no grant number available), the Swedish Diabetes Association Research Foundation(grant DIA2018-326), Swedish Research Council for Health, Working Life and Welfare-FORTE (grant STYA-2015/0003), Swedish Research Council (grant 2015–02503), the Institute of Health and Care Sciences of the University of Gothenburg and the Gothenburg University Centre for Person-centred care (GPCC).

**Competing interests** None declared.

**Patient and public involvement** Patients and/or the public were involved in the design, or conduct, or reporting, or dissemination plans of this research. Refer to the Methods section for further details.

**Patient consent for publication** Not required.

**Provenance and peer review** Not commissioned; externally peer reviewed.

**ORCID iD**
Anna Lena Brorsson http://orcid.org/0000-0002-8136-6340

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
