## [Reviewer comments · BMJ Open]

ARTICLE DETAILS

TITLE (PROVISIONAL)	A randomized controlled trial of a person-centred transition program for adolescents with type 1 diabetes (STEPSTONES-DIAB) – a study protocol
AUTHORS	Brorsson, Anna Lena; Bratt, Ewa-Lena; Moons, Philip; Ek, Anna; Jelleryd, Elisabeth; Torbjörnsdotter, Torun; Sparud-Lundin, Carina

VERSION 1 – REVIEW

REVIEWER	Dr Susan Sullivan-Bolyai University of Massachusetts Medical School-Graduate School of Nursing Worcester MA USA
REVIEW RETURNED	13-Jan-2020

GENERAL COMMENTS	Nice protocol but a few questions and missing information that would be beneficial:  *What is the theoretical framework that guides your study and links to your measures and outcomes? Please briefly describe to underscore how the theory guides your intervention *Consider a table to explicitly describe your intervention fidelity and monitoring plan: include the IF plan you are using including design, training, treatment delivery, treatment receipt and treatment enactment; what data will be reviewed for each category and time frame for each. *It would be helpful to share how you accounted statistically for retention/lost to f/u with your sample *Table the measures you are going to use with psychometrics for each measure *Have you considered qualitative interviews with your participants after they complete the trial to explore the context of the intervention?
--

REVIEWER	Ingo Menrath Luebeck University Department of Pediatrics Germany
REVIEW RETURNED	29-Jan-2020

GENERAL COMMENTS	Ad 4) Are the methods described sufficiently to allow the study to be repeated? The intended intervention is well described. However, the usual care remains a little bit unclear. How many centers will be involved? On page 12, line 23 the authors mention “In both centres...” Line 27 begins with “In one of the intervention centers,...”The next sentence starts with “In the other two centers,... It is not clear, how many
--

	centers will be involved in each study arm. What does it mean that in one of the intervention centers, no targeted transitional care interventions will be implemented? I thought that the intervention will be a targeted transitional care intervention. I understand that in two centers usual care comprises a somehow structured transition. The paper would benefit from a clearer distinction between usual care and the intervention. What is the job of midwives in the youth clinic? (page 13, line 14) The primary outcome is empowerment. It will be measured at all three measurements (T0, T1, T2). On page 15, line 29 it seems that empowerment will be analyzed at T2. How are changes from T0 to T2 considered (for example concerning the secondary outcomes there will be a analyzation of the changes form baseline to T2). Ad 15) The manuscript needs careful language editing. Here some examples: Page 17, line 25: The Impact scale describes emotional reactions to needing to control everyday life and to restricting one's diet. Page 22, line 25: Personal data regarding patients' health will be collected, and patients given the chance to be exposed to a new intervention that is different from current usual care. Page 25, line 3: A challenge in complex interventions like transition programs is the multiple interacting components and often challenging behaviours required by those delivering and receiving the intervention. Page 25, line 29 Some limitations need to be acknowledged in this study. Firstly, this intervention focuses on transition to adulthood, the actual transfer to adult care and, to a lesser extent, changes in the adult care services. Additional interventions may be needed to secure continuity of care after transfer. => The English is unusual. In addition, to my understanding this is not a limitation of the study.
--	--

VERSION 1 – AUTHOR RESPONSE

Reviewer: 1, Dr Susan Sullivan-Bolyai	Our response
What is the theoretical framework that guides your study and links to your measures and outcomes? Please briefly describe to underscore how the theory guides your intervention	We have tried to briefly clarify the theoretical assumption in the end of the introduction (see highlights in yellow). We have also described how measures and outcomes are linked to the theoretical framework in the method section (see highlights in yellow under heading Outcome measure). We refer to an article on intervention mapping (Acuna Mora, 2020) where we more extensively have elaborated on how theories have guided development of the transition program.
Consider a table to explicitly describe	We think we have described some of the aspects

your intervention fidelity and monitoring plan: include the IF plan you are using including design, training, treatment delivery, treatment receipt and treatment enactment; what data will be reviewed for each category and time frame for each.	of intervention fidelity and monitoring plan in figure 1 and table 1. The training of the TCs are described in the text (under the heading Intervention fidelity). However, we agree that treatment receipt and treatment enactment are insufficiently described and we have elaborated on how to monitor and document these aspects, see highlights in yellow.
It would be helpful to share how you accounted statistically for retention/lost to f/u with your sample.	The power calculation suggested 63 subjects in each RCT arm. Based on previous experiences, we assume a 10% drop-out rate (described under the heading Proposed sample size)
Table the measures you are going to use with psychometrics for each measure	We have now constructed such a table, perhaps it can be provided as a supplemental file depending on what editor suggests.
Have you considered qualitative interviews with your participants after they complete the trial to explore the context of the intervention?	Yes, qualitative interviews will be part of a PhD project and information is now included, see highlights in yellow (under the heading Intervention fidelity).
Reviewer: 2, Ingo Menrath	
The intended intervention is well described. However, the usual care remains a little bit unclear. How many centers will be involved? On page 12, line 23 the authors mention "In both centres..." Line 27 begins with "In one of the intervention centers,..." The next sentence starts with "In the other two centers,..." It is not clear, how many centers will be involved in each study arm. What does it mean that in one of the intervention centers, no targeted transitional care interventions will be implemented? I thought that the intervention will be a targeted transitional care intervention. I understand that in two centers usual care comprises a somehow structured transition. The paper would benefit from a clearer distinction between usual care and the intervention.	We agree that this was not completely clear and have now tried to clarify the description of usual care, see highlights in yellow.
What is the job of midwives in the youth clinic? (page 13, line 14)	We have tried to clarify this, see highlights in yellow.
The primary outcome is empowerment. It will be measured at all three measurements (T0, T1, T2). On page 15, line 29 it seems that empowerment will be analyzed at T2. How are changes from T0 to T2 considered (for example concerning the secondary outcomes there will be a analysis of the changes from baseline to T2).	Thank you for notifying that we had not explicitly mentioned that empowerment will also be analysed for change over time, as a secondary outcome. We have inserted information under the heading Secondary outcomes (highlighted in yellow).
Ad 15) The manuscript needs careful language editing.	The manuscript has undergone professional language editing. However, we agree that these examples need to be reformulated, we have revised the sentences

Here some examples: Page 17, line 25: The Impact scale describes emotional reactions to needing to control everyday life and to restricting one's diet. Page 22, line 25: Personal data regarding patients' health will be collected, and patients given the chance to be exposed to a new intervention that is different from current usual care. Page 25, line 3: A challenge in complex interventions like transition programs is the multiple interacting components and often challenging behaviours required by those delivering and receiving the intervention. Page 25, line 29 Some limitations need to be acknowledged in this study. Firstly, this intervention focuses on transition to adulthood, the actual transfer to adult care and, to a lesser extent, changes in the adult care services. Additional interventions may be needed to secure continuity of care after transfer. => The English is unusual. In addition, to my understanding this is not a limitation of the study.	accordingly.
--	---------------------

VERSION 2 – REVIEW

REVIEWER	Susan Sullivan-Bolyai UMMS-GSN Worcester MA USA
REVIEW RETURNED	28-Feb-2020

GENERAL COMMENTS	The previous critique concerns/comments were addressed in this resubmission.
--

REVIEWER	Ingo Menrath Luebeck University Dept. of Pediatrics Germany
REVIEW RETURNED	15-Mar-2020

GENERAL COMMENTS	Thank you for the opportunity to review the revision of the manuscript. The authors responded to all my comments.
---